# A PREDICTIVE CODING MODEL OF HIPPOCAMPO-NEOCORTICAL INTERACTIONS INVOLVED IN MEMORY REPLAY

## ABSTRACT

The neocortex and the hippocampus are two complementary learning systems which interact during memory construction and consolidation. The hippocampus stores episodic memories coming from the neocortex passing through the entorhinal cortex, and later replays them back to the neocortex to transform them into semantic memory during memory consolidation. It is thought that memory replay is a generative process, involved in imagining, because new episodes can also be generated and instantiated in the neocortex. Here we present a computational model of hippocampal-neocortical interactions based on a predictive coding network with two hidden layers, which are mapped onto the visual cortex and the entorhinal cortex. Improving on a previous implementation of this network, our simulations provide a mechanistic account of memory replay in the neocortex.

## 1 INTRODUCTION

According to the complementary learning systems (CLS) theory, the necortex is responsible for semantic memory, that is the general knowledge that we have about the world, whereas hippocampus stores episodic memories, which correspond to an individual's emotional and sensory experiences (Kumaran et al., 2016). For example, the experience of encountering a particularly odd-looking dog (his look, bark, smell and the surprise you felt when seeing it) can be stored as an episodic memory whereas the knowledge about what characterizes a typical dog is semantic memory.

After storage in the hippocampus, an episodic memory can be recalled from a corrupted version of it. Furthermore, episodic memories are replayed during rest or sleep for memory consolidation. Memory replay corresponds to the spontaneous reactivation of the activity corresponding to an episodic memory in the hippocampus (hippocampal replay) and its subsequent reinstantiation in the neocortex (cortical replay), so that it can gradually be integrated in the semantic memory of the neocortex. This idea is supported by empirical evidence from rodent studies during spatial navigation, where it was found that the rodent hippocampus generates sequences of activations during wakeful rest or sleep that reflect past trajectories (Buzsáki, 2015). In machine learning, experience replay has been shown to prevent catastrophic forgetting in a continual learning setting, where the learning of new tasks interferes with the knowledge of previously learned tasks. It consists of continually storing episodes in a memory buffer and replaying them when learning a new task.

It was later found that the internally generated hippocampal sequences are not merely replays of past trajectories, but also include paths that were never experienced before (Kumaran et al., 2016). This has prompted researchers from computational neuroscience and brain-inspired machine learning to hypothesize that the hippocampus is a generative model and that memory replay is a generative process, refered to as generative replay (Stoianov et al., 2022; van de Ven et al., 2020). Moreover, generative replay has been shown to improve the performance of reinforcement learning agents over experience replay Wang et al. (2025).

Despite the importance of memory replay for continual learning, its implementation in the neocortex and the hippocampus is not well understood. Because of the similarity of architecture in all its areas, researchers hypothesized that a common algorithm underlies computations in the neocortex (Friston, 2003; Hawkins et al., 2019). Principles of organization in the neocortex have emerged from empirical studies in the visual cortex, which have shown that this region is arranged hierarchically,

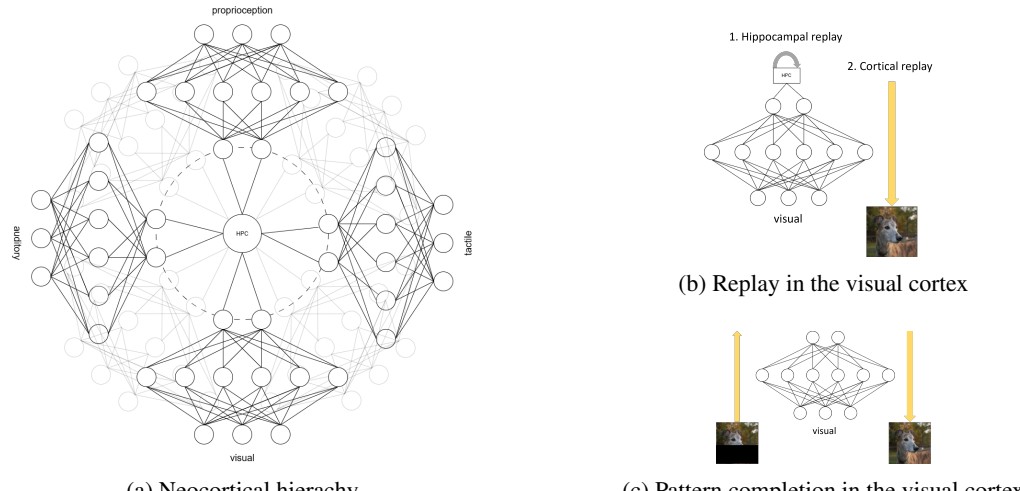

(a) Neocortical hierachy    (b) Replay in the visual cortex

(c) Pattern completion in the visual cortex

Figure 1: (a) The high-level representations in neocortical areas corresponding to different sensory modalities are combined in the entorhinal cortex (represented as the dotted circle) which is the input of the hippocampus (HPC), located at the apex of the neocortical hierarchy. (b) After storage of an episodic memory in the hippocampus, it can be replayed in the hippocampus (hippocampal replay) before being reinstantiated in the neocortex, including the visual cortex (cortical replay). (c) The visual cortex can complete an image based on semantic memory. In this illustration, the completion performance is perfect, as the recalled image is exactly the original image (as in Salvatori et al. (2021)), but it is not the case in reality (as explained in the main text).

with forward connections from lower to higher areas, and backward connections from higher to lower areas (Friston, 2003). At the apex of the neocortical hierarchy, the hippocampus receives input from the entorhinal cortex, which combines representations from different high-level neocortical areas of different sensory modalities (Barron et al., 2020), as illustrated in Figure 1a. Therefore, hippocampal replay could drive neocortical activity using backward connections from the entorhinal cortex to the neocortical hierachy, as illustrated for the visual cortex in Figure 1b.

Rooted in studies of the visual cortex, predictive coding has been proposed in computational neuroscience by Rao & Ballard (1999), and later extended by Friston (2003) as a general theory of cortical computation, which maps well to the neocortex in terms of architecture and information processing. Recently, Fontaine & Alexandre (2025b) investigated the role of the neocortex in semantic and episodic memory using a predictive coding network (PCN). They reproduced the result of Salvatori et al. (2021) that PCNs can store training images as memories (as illustrated in 1c), but showed that this is done by overfitting the network to a few training images. When the network is trained on more images, it generalizes better and is able to complete corrupted versions of training images based on semantic memory, but without recalling the details of the specific training images, supporting the CLS view that the neocortex is responsible for semantic memory. Even though the neocortex might not be responsible for episodic memory like the hippocampus, it supports episodic memory by allowing the episodic memories replayed by the hippocampus to be reinstantiated during cortical replay. Fontaine & Alexandre (2025b) modelled experience replay in a PCN, using two classes of MNIST digits, but the replayed images were found to be blurry, and the representations at the top of the hierarchy were found to be overlapping at the boundary of the two classes. In this paper, we tackle these limitations by accurately tuning the number of hidden units and stabilizing the convergence of the model on the full MNIST dataset using a learning rate scheduler. We found that adding more neurons in the second hidden layer allows the experience replays to be more accurate, by increasing the linear separability of representations corresponding to different classes at the top of the hierarchy, but at the cost of reconstruction and pattern completion performance. Furthermore, we extend the model to generative replay, by proposing a generative form of hippocampal replay.

## 2 RELATED WORKS

**Models of memory replay.** Other works captured the computation of experience replay and generative replay using deep learning models, but without being faithful to the architecture and information processing of the corresponding structures in the brain. (Spens & Burgess, 2024) proposed a model of memory consolidation consisting of experience replay, in which the neocortex is modelled as a VAE, whereas the hippocampus is modelled as a Modern Hopfield Network (Krotov & Hopfield, 2016) which replays episodic memories to the input of the VAE. In Stoianov et al. (2022), the hippocampus itself is modelled as a hierarchical generative model supporting generative replay.

**PCNs as models of the neocortex.** Since the work of Friston (2003), various papers investigated the idea that predictive coding could underly information processing in the neocortex. Brucklacher et al. (2023) proposed to study the representations learned by a PCN with two hidden layers and showed that the representations in the highest area are object-invariant when trained on sequences of continuously transformed images. In addition, they show that top-down reconstruction of inputs from latent variables when blanking out the input becomes less accurate in higher areas, suggesting that higher areas encode reduced information such as object identity. In our model, memory replay is modelled without blanking out the input, in a biologically plausible manner, as memory replay can occur while the brain is exposed to sensory input. Salvatori et al. (2021) showed that PCNs outperform other models in auto-associative memory (AM) and suggested based on Barron et al. (2020) that the top layer of their PCN could correspond to the hippocampus. Tang et al. (2023) extended their model by adding a recurrent one-layer PCN to the top of a hierarchical PCN modelling the neocortex and Li et al. (2025) showed that PCNs also detect novelty at different levels of abstraction in the hierarchy. The architecture and the inference and learning rules of our model are similar to that of Salvatori et al. (2021) as both models are based on the model by Friston (2003). The main difference is that our work models memory replay and not auto-associative memory, which is a function associated to the hippocampus rather than the neocortex according to Fontaine & Alexandre (2025b). However, the pattern completion performance of our model will be evaluated on unseen images, to tune the size of the top level. Indeed, even though the neocortex likely cannot recall details of episodic memories, it can still complete corrupted patterns based on semantic memory. Another difference with this line of work is that the top level of our PCN corresponds to the entorhinal cortex, and not the hippocampus. Indeed, as the hippocampus is able to perform one-shot storage of episodic memories, we believe that it cannot be fully modelled using a PCN.

**Image generation in PCN.** Other works studied the generation of inputs in predictive coding networks (PCN), but without specifying how it can be implemented in the neocortex in the presence of sensory inputs, or its relation to the hippocampus. On the one hand, Oliviers et al. (2024) proposed Monte Carlo predictive coding for learning probability distributions of sensory inputs, arguing that classical predictive coding demonstrated limited performance in generative tasks. On the other hand, preliminary work by Millidge (2019) showed that a PCN with one hidden layer can be used to generate inputs by sampling points close to the training data in the latent space, even though the generated samples were blurry and the authors did not describe their method for sampling. Ororbia & Kifer (2022) proposed to generate images from an extended version of PCN with ancestral sampling, but only three images per class are shown. In addition, using the model proposed by Friston (2003), the authors only show nearest neighbor samples that match an original data point for each class, leaving aside a large part of the image space which is covered by the generative model.

## 3 PREDICTIVE CODING

Predictive coding networks (PCN) are based on hierarchical generative models with $L$ layers, of the type

$$\forall l \in \{0, 1, ..., L-1\}, p_{\boldsymbol{\theta}_l}(\boldsymbol{h_l} \mid \boldsymbol{h_{l+1}}) = \mathcal{N}(\boldsymbol{h_l}; \boldsymbol{\mu}_l, \boldsymbol{\Sigma}_l), \boldsymbol{\mu}_l = f_l(\boldsymbol{\mu}_{l+1}; \boldsymbol{\theta}_l)$$

$$p_{\boldsymbol{\theta}_L}(\boldsymbol{h_L}) = \mathcal{N}(\boldsymbol{h_L}; \boldsymbol{\mu_L}, \boldsymbol{\Sigma_L}), \boldsymbol{\mu_L} = \boldsymbol{\theta_L}$$

where $\boldsymbol{h} = (\boldsymbol{h_0}, \boldsymbol{h_1}, ..., \boldsymbol{h_L})$ are the states and $\boldsymbol{\theta} = (\boldsymbol{\theta_0}, \boldsymbol{\theta_1}, ..., \boldsymbol{\theta_L})$ are the parameters. Level 0 is the input level, so $\boldsymbol{h_0}$ is the input state and $\boldsymbol{h_1}, \boldsymbol{h_2}, ..., \boldsymbol{h_L}$ are the latent states. In practice, we use $\boldsymbol{\mu_L} = \boldsymbol{0}$ and $\boldsymbol{\Sigma}_l = \boldsymbol{I}$ for all $l \in \{0, 1, ..., L\}$. Therefore, the prior on the latent state of level $L$ is a centered isotropic multivariate Gaussian $p_{\boldsymbol{\theta}_L}(\boldsymbol{h_L}) = \mathcal{N}(\boldsymbol{h_L}; \boldsymbol{0}, \boldsymbol{I})$ with no parameter.

Recognition is assumed to be deterministic, such that for an input $\boldsymbol{x}$,

$$q_{\boldsymbol{\phi}}(\boldsymbol{h_1}, \boldsymbol{h_2}, ..., \boldsymbol{h_L} \mid \boldsymbol{x}) = \prod_{l=1}^{L} \delta(\boldsymbol{h_l} - \boldsymbol{\phi_l})$$

where $\boldsymbol{\phi} = (\boldsymbol{\phi_1}, \boldsymbol{\phi_2}, ..., \boldsymbol{\phi_L})$ is an estimate of the latent states $\boldsymbol{h_1}, \boldsymbol{h_2}, ..., \boldsymbol{h_L}$ corresponding to the input $\boldsymbol{x}$.

Inference of the latent states for an input $\boldsymbol{x}$ results from the minimization of a lower bound to the negative likelihood, called variational free energy

$$\mathcal{L}(\boldsymbol{\theta}, \boldsymbol{\phi}; \boldsymbol{x}) = -\mathbb{E}_{q_{\boldsymbol{\phi}}(\boldsymbol{h_1}, \boldsymbol{h_2}, ..., \boldsymbol{h_L} \mid \boldsymbol{x})} \left[ \log p_{\boldsymbol{\theta}}(\boldsymbol{x}, \boldsymbol{h_1}, \boldsymbol{h_2}, ..., \boldsymbol{h_L}) \right]$$

$$= -\log p_{\boldsymbol{\theta}}(\boldsymbol{x}, \boldsymbol{\phi_1}, \boldsymbol{\phi_2}, ..., \boldsymbol{\phi_L})$$

$$= -\log p_{\boldsymbol{\theta_0}}(\boldsymbol{x} \mid \boldsymbol{\phi_1}) - \log p_{\boldsymbol{\theta_1}}(\boldsymbol{\phi_1} \mid \boldsymbol{\phi_2}) - ... - \log p_{\boldsymbol{\theta_{L-1}}}(\boldsymbol{\phi_{L-1}} \mid \boldsymbol{\phi_L}) - \log p(\boldsymbol{\phi_L})$$

$$= \frac{1}{2} \sum_{l=0}^{L-1} \left[ \boldsymbol{\xi}_l^T \boldsymbol{\xi}_l + \log |\boldsymbol{\Sigma}_l| \right] - \log p(\boldsymbol{\phi_0}) + \text{constant}$$

where taking the logarithm of a Gaussian distribution results in a quantity

$$\boldsymbol{\xi}_l = \boldsymbol{\Sigma}_l^{-\frac{1}{2}}(\boldsymbol{\phi}_l - f_l(\boldsymbol{\phi}_{l+1}; \boldsymbol{\theta}_l)). \tag{1}$$

which can be seen as a prediction error for layer $l$. Thus, the variational free energy corresponds to the sum of prediction errors in all layers, and PCNs learn hierarchical predictive representations of the input.

When presented with an input $\boldsymbol{x}$, the latent states are updated according to

$$\dot{\boldsymbol{\phi}}_l = -\nabla_{\boldsymbol{\phi}_l} \mathcal{L}(\boldsymbol{\theta}, \boldsymbol{\phi}; \boldsymbol{x}) = -\frac{\partial \boldsymbol{\xi}_{l-1}^T}{\partial \boldsymbol{\phi}_l} \boldsymbol{\xi}_{l-1} - \frac{\partial \boldsymbol{\xi}_l^T}{\partial \boldsymbol{\phi}_l} \boldsymbol{\xi}_l \tag{2}$$

until the variational free energy is minimized. In practice, we only update the latent states $T$ times during training.

Similarly, learning of the parameters $\boldsymbol{\theta} = (\boldsymbol{\theta_0}, \boldsymbol{\theta_2}, ..., \boldsymbol{\theta_{L-1}})$ corresponds to the minimization of the variational free energy $F(\boldsymbol{\theta}) = \mathbb{E}_{p(\boldsymbol{x})}[\mathcal{L}(\boldsymbol{\theta}, \boldsymbol{\phi}; \boldsymbol{x}))]$. After several image presentations, the parameters are updated once following

$$\forall l \in \{1, 2, ..., L\}, \dot{\boldsymbol{\theta}}_l = -\nabla_{\boldsymbol{\theta}_l} F = -\mathbb{E}_{p(\boldsymbol{x})} \left[ \frac{\partial \boldsymbol{\xi}_l^T}{\partial \boldsymbol{\theta}_l} \boldsymbol{\xi}_l \right]. \tag{3}$$

This algorithm can be implemented in a neural network, hence the name PCN, with only local computations for inference and learning. In a PCN, each level $l$ consists of two types of neurons, with activity $\boldsymbol{\phi}_l$ and $\boldsymbol{\xi}_l$ respectively. When mapped to the neocortical hierarchy, level $l+1$ is the level above $l$, with level 0 at the bottom and level $L$ at the top. From equation 1, it can be seen that neurons $\boldsymbol{\xi}_l$ compute the prediction errors, based on lateral connections with neurons $\boldsymbol{\phi}_l$ at the same level and inhibitory feedback connections with neurons $\boldsymbol{\phi}_{l+1}$ in the level above, which provide the predictions. Equation 2 shows that neurons $\boldsymbol{\phi}_l$ receive connections from error neurons in the same level $\boldsymbol{\xi}_l$ and the level below $\boldsymbol{\xi}_{l-1}$. In addition, it can be seen that equation 3 corresponds to Hebbian learning, as shown in the next section.

While it is standard in the predictive coding literature to use fixed covariances $\boldsymbol{\Sigma}_l = \boldsymbol{I}$, it was proposed that the inverse covariance, called precision, $\boldsymbol{\Sigma}_l^{-1}$ is predicted by higher layers and mediates attention (Feldman & Friston, 2010). Indeed, if precision $\boldsymbol{\Sigma}_l^{-1}$ is low, prediction error $\boldsymbol{\xi}_l = \boldsymbol{\Sigma}_l^{-\frac{1}{2}}(\boldsymbol{\phi}_l - f_l(\boldsymbol{\phi}_{l+1}; \boldsymbol{\theta}_l))$ is low, and will not influence the update of neuron activities and weights. As explained by Li (2023) based on Clark (2016), the precision parameter controls the degree to which the brain attends to the external input or to the internal prediction, determining whether it is performing perception or imagination.

## 4 METHODS

Building upon Fontaine & Alexandre (2025a), we propose a predictive coding model of the visual cortex and show that it learns hierarchical predictive representations of MNIST images, that support memory replay.

## 4.1 MODEL

Our model is a PCN with $L = 2$ layers

$$p_{\boldsymbol{\theta_0}}(\boldsymbol{h_0} \mid \boldsymbol{h_1}) = \mathcal{N}(\boldsymbol{h_0}; \boldsymbol{W_0 h_1}, \boldsymbol{I})$$
$$p_{\boldsymbol{\theta_1}}(\boldsymbol{h_1} \mid \boldsymbol{h_2}) = \mathcal{N}(\boldsymbol{h_1}; f(\boldsymbol{W_1 h_2} + \boldsymbol{b_1}), \boldsymbol{I})$$
$$p(\boldsymbol{h_2}) = \mathcal{N}(\boldsymbol{h_2}; \boldsymbol{0}, \boldsymbol{I})$$

where $\boldsymbol{\theta}_0 = \boldsymbol{W_0}$ and $\boldsymbol{\theta}_1 = (\boldsymbol{W_1}, \boldsymbol{b_1})$. Indeed, the bias and non-linearity are not required in the input layer, because an image can be represented as a linear combination of basis functions (Olshausen & Field, 1996).

The variational free energy for an input $\boldsymbol{x}$ is

$$\mathcal{L}(\boldsymbol{\theta}, \boldsymbol{\phi}; \boldsymbol{x}) = \frac{1}{2}\boldsymbol{\xi}_0^\top \boldsymbol{\xi}_0 + \frac{1}{2}\boldsymbol{\xi}_1^\top \boldsymbol{\xi}_1 + \frac{1}{2}\boldsymbol{\xi}_2^\top \boldsymbol{\xi}_2 + \text{constant}$$

where

$$\boldsymbol{\xi}_0 = \boldsymbol{x} - \boldsymbol{W_0 \phi}_1 \tag{4}$$
$$\boldsymbol{\xi}_1 = \boldsymbol{\phi}_1 - f(\boldsymbol{W_1 \phi}_2 + \boldsymbol{b_1}) \tag{5}$$
$$\boldsymbol{\xi}_2 = \boldsymbol{\phi}_2. \tag{6}$$

## 4.2 TRAINING ALGORITHM

Let us consider a dataset $\boldsymbol{X} = \{\boldsymbol{x}^{(i)}\}_{i=1}^N$ of $N$ i.i.d. samples of a continous variable $\boldsymbol{x}$. The log likelihood can be written $\log p_{\boldsymbol{\theta}}(\boldsymbol{x^{(1)}}, ..., \boldsymbol{x^{(N)}}) = \sum_{i=1}^N \log p_{\theta}(\boldsymbol{x}^{(i)})$. Therefore, the variational free energy of the dataset $\boldsymbol{X}$ is

$$\mathcal{L}(\boldsymbol{\theta}; \boldsymbol{X}) = \sum_{i=1}^N \mathcal{L}(\boldsymbol{\theta}, \boldsymbol{\phi}; \boldsymbol{x^{(i)}}),$$

which can be estimated based on minibatches

$$\mathcal{L}(\boldsymbol{\theta}; \boldsymbol{X}) \approx \mathcal{L}^M(\boldsymbol{\theta}; \boldsymbol{X}^M) = \frac{N}{M}\sum_{i=1}^M \mathcal{L}(\boldsymbol{\theta}, \boldsymbol{\phi}^{(i)}; \boldsymbol{x}^{(i)}) \tag{7}$$

where the minibatch $\boldsymbol{X}^M = \{\boldsymbol{x}^{(i)}\}_{i=1}^M$ is randomly drawn from the dataset $\boldsymbol{X}$.

Given a minibatch $\boldsymbol{X}^M$, the latent states $\boldsymbol{\phi}_0^{(i)}$ and $\boldsymbol{\phi}_1^{(i)}$ are updated during $T$ iterations for each datapoint $\boldsymbol{x^{(i)}}$ to minimize the variational free energy $\mathcal{L}(\boldsymbol{\theta}, \boldsymbol{\phi}^{(i)}; \boldsymbol{x}^{(i)})$. The update rules for the two layers in our model are

$$\Delta \boldsymbol{\phi}_1 = \alpha(\boldsymbol{W_0}^\top \boldsymbol{\xi}_0 - \boldsymbol{\xi}_1) \tag{8}$$
$$\Delta \boldsymbol{\phi}_2 = \alpha(\boldsymbol{W_1}^\top \text{diag}\left[f'(\boldsymbol{W_1 \phi}_2 + \boldsymbol{b_1})\right]\boldsymbol{\xi}_1 - \boldsymbol{\xi}_2). \tag{9}$$

where $\alpha$ is the inference rate.

Then, the parameters $\boldsymbol{\theta}_0 = \boldsymbol{W_0}$ and $\boldsymbol{\theta}_1 = (\boldsymbol{W_1}, \boldsymbol{b_1})$ are updated once to minimize the estimate $\mathcal{L}^M(\boldsymbol{\theta}; \boldsymbol{X}^M)$ given in equation 7. Thus, the learning rules can be calculated from the sum of gradients $\sum_{i=1}^M \nabla_{\boldsymbol{\theta}_l} \mathcal{L}(\boldsymbol{\theta}; \boldsymbol{x}^{(i)})$

$$\Delta \boldsymbol{W_0} = \beta \sum_{i=1}^M \boldsymbol{\xi_0}^{(i)} \boldsymbol{\phi_1}^{(i)\top}$$

$$\Delta \boldsymbol{W_1} = \beta \sum_{i=1}^M \left[\boldsymbol{\xi_1}^{(i)} \odot f'(\boldsymbol{W_1 \phi_2}^{(i)} + \boldsymbol{b_1})\right] \boldsymbol{\phi_2}^{(i)\top}$$

$$\Delta \boldsymbol{b_1} = \beta \sum_{i=1}^M \boldsymbol{\xi_1}^{(i)} \odot f'(\boldsymbol{W_1 \phi_2}^{(i)} + \boldsymbol{b_1}).$$

where $\odot$ is the element-wise product and $\beta$ is the learning rate. We train the model on a dataset $\boldsymbol{X}$ for multiple epochs, and reduce the learning rate by a multiplicative factor $\gamma < 1$ at each epoch of training to prevent instability issues. Details can be found in section A.1 of the appendix.

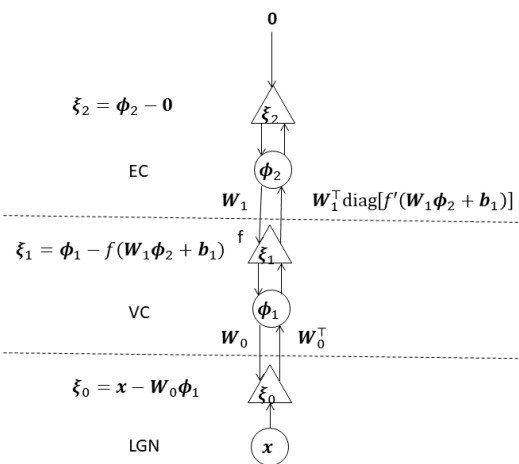

Figure 2: PCN with 3 levels mapped to lateral geniculate nucleus (LGN), visual cortex (VC) and entorhinal cortex (EC). Each node corresponds to multiple neurons. Circle nodes correspond to units $\phi_l$ and triangle nodes correspond to error units $\xi_l$. Each connection between nodes corresponds to a fully connected network, with excitatory connections in the upward direction and inhibitory connections in the downward direction. We consider the upmost prediction to be 0, to have a standard normal distribution as prior.

### 4.3 MAPPING TO THE BRAIN

Figure 2 shows how this algorithm can be implemented in a neural network mapped onto the visual pathway of the brain, with only local computations. The three levels in the network correspond, from bottom to top, to the lateral geniculate nucleus (LGN) in thalamus (and not the retina because it doesn't receive feedback connections from LGN), the visual cortex (VC) and the entorhinal cortex (EC). This mapping allows us to study memory replay after training the model with algorithm 1.

At inference time, the latent states are updated until convergence or until a maximum number of iterations $T_{\max}$ is reached in order to study the converged representations. Convergence occurs when the relative change in the norm of the latent state $\phi_l$ is smaller than a threshold $\epsilon$, i.e.

$$\frac{\|\nabla_{\phi_l}\mathcal{L}(\boldsymbol{\theta}, \boldsymbol{\phi}; \boldsymbol{x})\|}{\|\phi_l\|} < \epsilon.$$

During perception, the network is driven by the input image in LGN. While the LGN is set to the image, the VC and EC converge to hierarchical predictive representations of the image following the inference rules 8 and 9 respectively. The representation in VC is predictive of the image, as the prediction $\boldsymbol{W_0}\phi_1$ is a reconstruction of the image. The representations in EC can be stored by the hippocampus (not explicitly modelled) and later replayed during experience replay.

During memory replay, hippocampal replay first outputs the EC representation of an image stored by the hippocampus in the case of experience replay or a sample generated in the latent space of EC in the case of generative replay (as described in the next paragraph). Then, during cortical replay, the network is driven by the representation in EC obtained from hippocampal replay. While setting the EC layer to the corresponding representation, the VC converges to the replayed representation following the inference rule 8. As the network can also be presented with an input in LGN during memory replay, the representation in VC is protected from ascending input in LGN by setting the precision $\boldsymbol{\Sigma_0^{-1}}$ in the LGN to 0, preventing the prediction errors in LGN to influence the activity in VC. In this way, attention is focused on the representation in EC, and not on the current input. Then, the prediction $\boldsymbol{W_0}\phi_1$ based on the replayed representation $\phi_1$ in VC corresponds to the replayed image.

In generative replay, sampling of the latent space of EC is class-conditioned. Indeed, we fit a multivariate Gaussian distribution to each class in the latent space of EC, by estimating the mean and

covariance of training samples in each class. Then, samples from a given class can be generated by sampling the corresponding Gaussian distribution.

During perception, the network can also perform pattern completion when presented with a corrupted (noisy or incomplete) input. Following Salvatori et al. (2021), while the LGN layer is initialized to the corrupted input, the VC and EC converge following the same dynamics as in regular perception (equations 8 and 9), whereas the corrupted part of LGN converges following to the inference rule

$$\Delta \phi_0 = -\alpha \phi_0.$$

obtained by calculating the gradient of the variational free energy of the input with respect to the corrupted part of LGN. Then, the prediction $W_0 \phi_1$ based on the recalled representation $\phi_1$ in VC corresponds to the recalled image.

### 4.4 EXPERIMENTS

The model is trained on the MNIST dataset, which contains images of handwritten digits from 0 to 9. The original training set of 60,000 images is split into a training set of size 50,000 and a validation set of size 10,000. The validation set is used to evaluate the model during training and hyperparameter tuning. After training, we evaluate the model on the original test set of 10,000 images.

Hyperparameter values are chosen based on empirical trials informed by the predictive coding literature and summarized in Table 1 in the appendix. In addition, the number of hidden units in level 1 is obtained by minimizing the variational free energy on the validation set for a PCN with $L = 1$ using grid search, as shown in Figure 7 in the appendix. The choice of the number of hidden units in level 2 is more complex and results from a trade-off between different metrics.

In our simulations, we study the influence of the number of hidden units in level 2 on the learned representations and on memory replay, both quantitatively and qualitatively. Quantitatively, we evaluate the predictive performance of the model, the quality of the experience replays and the pattern completion performance using the reconstruction, replay and completion errors, based on the mean squared error (MSE). The MSE between two flattened images $x$ and $\hat{x}$ is

$$\text{MSE}(\boldsymbol{x}, \hat{\boldsymbol{x}}) = \frac{1}{N_{\text{pixels}}} \sum_{i=1}^{N_{\text{pixels}}} (x_i - \hat{x_i}).$$

The reconstruction and completion errors are computed between the original and reconstructed images and between the original and recalled images respectively, averaged over the validation set, whereas the replay error is computed between the original and replayed images, averaged over the training set. In addition, we evaluate the linear separability of the latent manifolds corresponding to the different classes in level 2 of the model using the classification accuracy of a simple multinomial logistic regression. On the qualitative side, we examine examples of reconstructions of images from the validation set, as well as examples of experience replay corresponding to images from the training set. Additionnally, we show examples of images generated by generative replay and visualize their hierarchical representations.

## 5 RESULTS

The reconstruction error, completion error, replay error and classification accuracy are plotted against the number of hidden units in level 2 in Figure 3. These measures are evaluated after training the model until convergence of all layers, as shown in Figure 8 in the appendix. One the one hand, the left plot of Figure 3 shows that the second hidden layer in our model does not improve the predictive power of its representations. On the contrary, the reconstruction error increases with the number of units in the second hidden level, and the same model with the top layer removed has a lower reconstruction error (shown as the blue dotted line) than any of the models with two hidden layers. Similarly, the completion error increases with the number of units in the second hidden level, but it is always lower than for a model with only one hidden layer, suggesting that adding a second layer is still beneficial to the completion performance, contrary to the reconstruction performance. On the other hand, the right plot of Figure 3 shows that the quality of replay and the linear separability of the classes in the second hidden level increases with the number of units in that level.

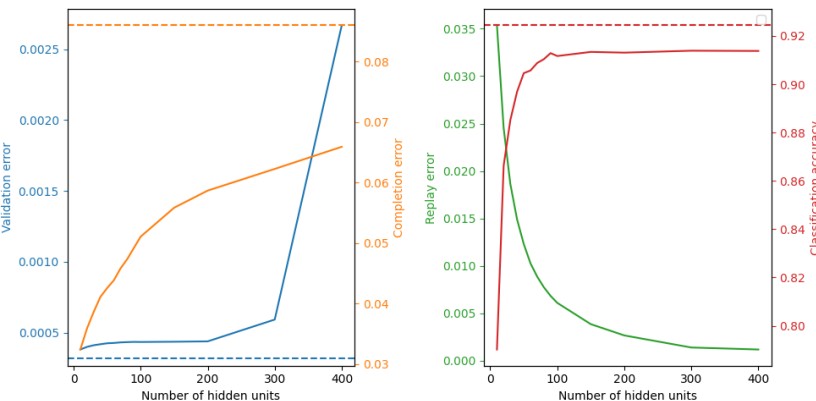

Figure 3: Quantitative evaluation of our model according to different metrics, depending on the number of hidden units in level 2. The dotted lines correspond to the same model with $L = 1$.



Figure 4: The same minibatch of images from the training set (shown in Observation) is replayed in PCNs with different widths $n_2$ at the top level.

Therefore, choosing the number of units in level 2 based on these metrics is not straightforward, and we will turn to the qualitative evaluation.

In Figure 4, we visualize examples of experience replay obtained with models of different widths. As shown in Fontaine & Alexandre (2025a), setting the width of the top level to the number of classes (i.e. $n_2 = 10$) results in replayed images that are blurry and that do not retain the details of the original images. This issue is solved by increasing the width of the top level to 30. Increasing it further to 100 improves the sharpness of the replayed images, but at the cost of the completion performance. Indeed, Figure 5 shows that while a width of 30 enables the network to semantically complete the bottom half of images taken from the validation set that were masked, a higher width of 100 makes the completion uninformative of the classes of the masked digits.

However, visual inspection of the reconstructions of images from the validation set and generative replays does not differentiate the models with different widths. Indeed, the difference in reconstruction errors between the different models is imperceptible in the reconstructed images. Similarly, the latent spaces and the quality of the images generated by replay are similar in the different models, despite the difference in replay error and classification accuracy. Thus, the trade-off between the replay fidelity and completion performance leads us to choose a width of 30.

Examples of reconstructions of images from the validation set and replayed images obtained by generative replay for a model with $n_2 = 30$ units in level 2 are shown in Figure 6. It can be seen from Figure 6a that the representations learned by the model are perfectly predictive of images it has never seen during training. In Figure 6c, we plotted hierarchical representations obtained with generative replay, over hierarchical representations inferred from images of the validation set. The representations of the different classes are well separated in all three levels, including the top level which was found to have overlapping clusters in Fontaine & Alexandre (2025a), and the representations generated by generative replay mostly fall within the right clusters in all levels. This is confirmed

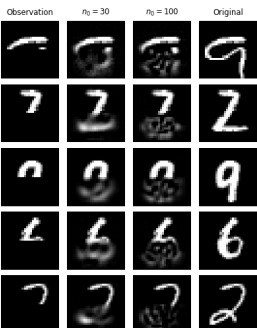

Figure 5: Five images from the validation set (shown in the Original column) were masked (as shown in the Observation column) and completed by a PCN with $n_2 \in \{30, 100\}$ neurons in the top level.

by looking at the generated images in Figure 6b. Most of the images generated for each class are realistic examples of their classes, even though some of them are blurry.

## 6 DISCUSSION

We proposed a model of hippocampo-neocortical interactions involved in memory replay using a PCN. Our work shows that predictive coding accounts for cortical replay, i.e. the reinstantiation in the neocortex of an episodic memory replayed or generated in the hippocampus. To this purpose, we modelled hippocampal replay in a minimal way, both in experience replay and generative replay, by mapping the top level of the PCN to the output of the hippocampus, the entorhinal cortex. Some of the images generated by our model were found to be out-of-distribution, probably because the simple Gaussian distribution we used does not capture the complex, non-linear geometry of the latent manifolds. This issue could be solved using a Riemannian metric (Arvanitidis et al., 2021). However, to provide a more complete account of memory replay, future work should aim at modelling the hippocampal formation with its different components to understand hippocampal replay mechanistically. In this way, realistic in-distribution samples will naturally be generated thanks to the learned connection between the entorhinal cortex and the hippocampus. In the hippocampus, generative replay should encompass experience replay as a generative process which samples both existing episodic memories and imagined ones.

Our work also contributes to understanding hierarchical representations in PCNs. It reveals on the one hand that adding more neurons in the top level of the network improves the fidelity of experience replay and the linear separability of the representations corresponding to different classes in the top level. Indeed, expanding the dimensionality of the activity space of patterns increases their separability (Cayco-Gajic & Silver, 2019), and the increased separability could lead to better replay, as there is less interference between patterns. However, our study also shows that the width of the top level is detrimental to the model's predictive and completion performance on unseen images. We also found that adding a second hidden layer is beneficial to pattern completion, but not to image reconstruction. The negative result about reconstruction can be interpreted in the light of efficient coding: any image can be described by the linear combination of a set of basis functions (Olshausen & Field, 1996). In our model, these basis functions correspond to the weights of the bottom layer, between VC and LGN, which enable for the prediction of images in LGN. As we have tuned the number of neurons in VC to minimize the reconstruction error on the validation set (see Figure 7 in the appendix), we have found such a set of basis functions for the MNIST dataset. Therefore, adding a second layer can only decrease the validation error. These results suggest that layers higher in the neocortical hierarchy have a role in pattern completion, but not in reconstruction. Furthermore, the work of (Brucklacher et al., 2023) and (Li et al., 2025) indicate that higher areas in PCNs encode object identity. However, we find that the classification accuracy of our model is higher if we remove the top hidden layer (shown as the red dotted line in Figure 3). The role of depth will be investigated in future work.

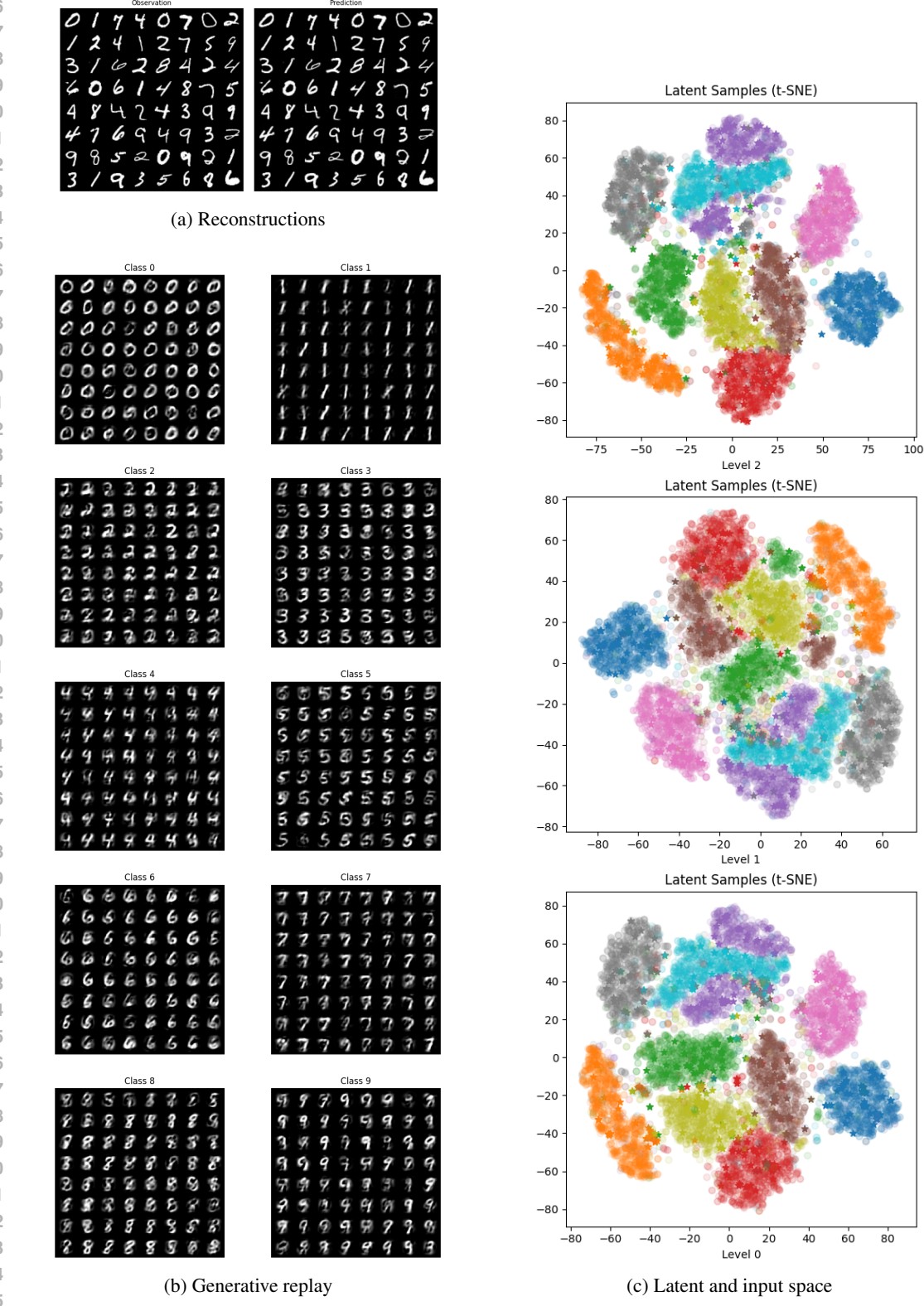

(a) Reconstructions

(b) Generative replay

(c) Latent and input space

Figure 6: Visualizations for a model with $n_2 = 30$ units in the top level. (a) Images reconstructed (right) by a PCN with $n_2 = 30$ units in the top level for a random mini-batch of images from the validation set (left). (b) Images generated by replay for each class. (c) Hierarchical representations of the images generated by generative replay (star-shaped markers) and of the images of the validation set (transparent circle markers), visualized in 2D using t-SNE. Each image is represented as one data point in each of the three subplots, colored according to the class.

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

## A  APPENDIX

### A.1  ALGORITHM

The algorithm described in section 4.2 is summarized in algorithm 1 where $\alpha, \beta$ are the inference and learning rates. The constant $\frac{N}{M}$ from equation 7 is factorized in the learning rate $\beta$. To prevent instabilities which occured systematically during training, we propose an exponential learning rate scheduler

$$\beta_{\text{epoch}} = \gamma \times \beta_{\text{epoch}-1}$$

which decays the learning rate $\beta$ by a multiplicative factor $\gamma$ at each epoch.

Initialization parameters include the standard deviations $\sigma_{\boldsymbol{W}}$ and $\sigma_{\boldsymbol{\phi}}$ and the number of dimensions of latent state $\boldsymbol{\phi_0}$.

### A.2  EXPERIMENTS

---

**Algorithm 1** Model training with minibatches

---

$\boldsymbol{W_1}, \boldsymbol{W_2} \leftarrow$ Sample from $\mathcal{N}(0, \sigma_{\boldsymbol{W}})$
$\boldsymbol{b_1} \leftarrow \mathcal{U}(-\frac{1}{n_0}, \frac{1}{n_0})$
**repeat**
    **for** $k = 1$ to $n_{\text{batches}}$ **do**
        $\boldsymbol{X}^M \leftarrow$ Random minibatch of $M$ datapoints drawn from $\boldsymbol{X}$
        **for** $i = 1$ to $M$ **do**
            $\phi_2^{(i)} \leftarrow \boldsymbol{x}^{(i)}$
            $\phi_0^{(i)}, \phi_1^{(i)} \leftarrow$ Sample from $\mathcal{N}(0, \sigma_{\boldsymbol{\phi}})$
            $\boldsymbol{\xi}_0^{(i)}, \boldsymbol{\xi}_1^{(i)}, \boldsymbol{\xi}_2^{(i)} \leftarrow$ Calculate the corresponding errors (equation 4)
            **for** $t = 1$ to $T$ **do**
                $\phi_1^{(i)} \leftarrow \phi_1^{(i)} + \alpha(\boldsymbol{W_2}^\top \boldsymbol{\xi}_2^{(i)} - \boldsymbol{\xi}_1^{(i)})$
                $\phi_0^{(i)} \leftarrow \phi_0^{(i)} + \alpha(\boldsymbol{W_1}^\top \text{diag}\left[f'(\boldsymbol{W_1}\phi_0^{(i)} + \boldsymbol{b_1})\right] \boldsymbol{\xi}_1^{(i)} - \boldsymbol{\xi}_0^{(i)})$
                $\boldsymbol{\xi}_0^{(i)}, \boldsymbol{\xi}_1^{(i)}, \boldsymbol{\xi}_2^{(i)} \leftarrow$ Calculate the corresponding errors (equation 4)
            **end for**
        **end for**
        $\boldsymbol{W_2} \leftarrow \boldsymbol{W_2} + \beta \sum_{i=1}^M \boldsymbol{\xi}_2^{(i)} \phi_1^{(i)\top}$
        $\boldsymbol{W_1} \leftarrow \boldsymbol{W_1} + \beta \sum_{i=1}^M \left[\boldsymbol{\xi}_1^{(i)} \odot f'(\boldsymbol{W_1}\phi_0^{(i)} + \boldsymbol{b_1})\right] \phi_0^{(i)\top}$
        $\boldsymbol{b_1} \leftarrow \boldsymbol{b_1} + \beta \sum_{i=1}^M \boldsymbol{\xi}_1^{(i)} \odot f'(\boldsymbol{W_1}\phi_0^{(i)} + \boldsymbol{b_1})$
    **end for**
**until** variational free energy $\mathcal{L}(\boldsymbol{\theta}; \boldsymbol{X})$ is minimized

---

Table 1: Hyperparameter values

| Parameter | Value |
|---|---|
| Activation function $f$ | $\tanh$ |
| Batch size | 64 |
| Standard deviation $\sigma_{\boldsymbol{W}}$ | 0.01 |
| Standard deviation $\sigma_{\boldsymbol{\phi}}$ | 0.05 |
| Number of iterations $T_{\text{train}}$ | 50 |
| Number of iterations $T_{\text{valid}}$ | 200 |
| Maximum number of iterations $T_{\text{max}}$ | 20000 |
| Convergence threshold $\epsilon$ | $2 \times 10^{-4}$ |
| Inference rate $\alpha$ | 0.01 |
| Inference optimizer | SGD |
| Initial learning rate $\beta_0$ | $10^{-5}$ |
| Learning rate decay factor $\gamma$ | 0.99 |
| Learning optimizer | Adam |

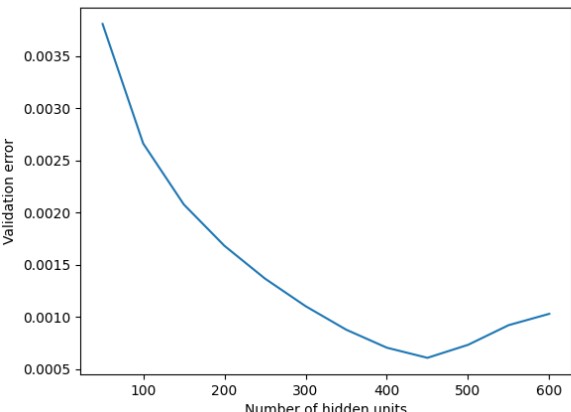

Figure 7: RMSE between original and reconstructed images averaged over the validation set for a PCN with $L = 1$.

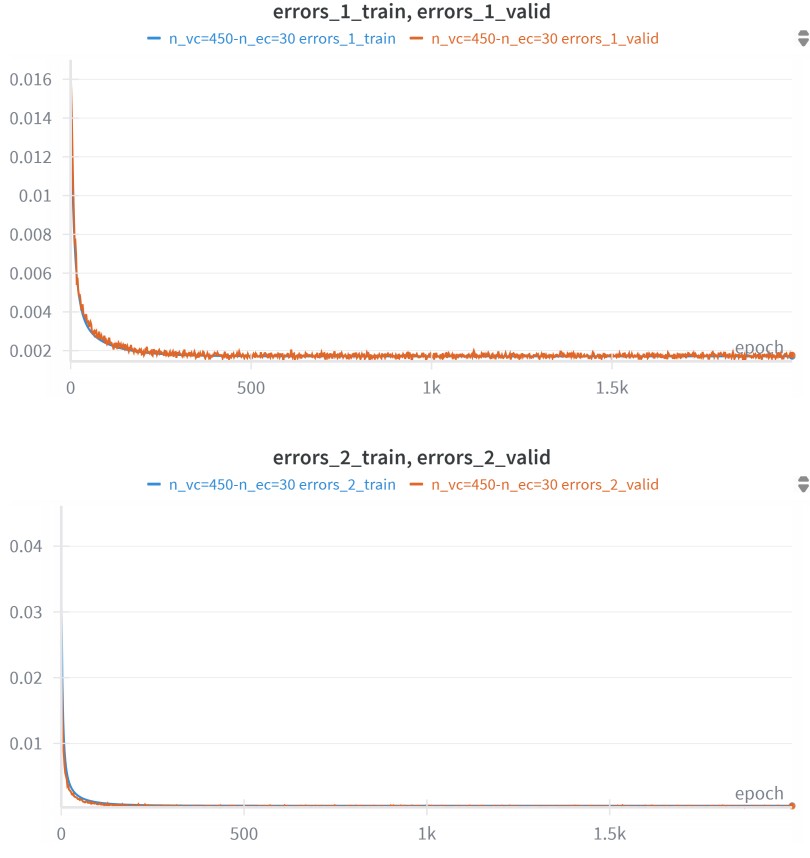

Figure 8: Mean prediction errors in level 1 and 2 of a model with 30 units in level 0, averaged over the whole training set (blue line) and a random minibatch from the validation set (orange line) at each epoch. The model converges after 2000 epochs thanks to the learning rate scheduler.