# OpenReview forum: "A predictive coding model of hippocampo-neocortical interactions involved in memory replay"
_ICLR.cc/2026/Conference — Submitted to ICLR 2026_

### Official Review · Reviewer_siHR · 2025-10-23

**Soundness:** 2
**Presentation:** 2
**Contribution:** 1
**Rating:** 2
**Confidence:** 5

**Summary:**

This paper proposes a 2-layered predictive coding model for neocortical reconstruction and replay. Based on theories and experimental observations that memory replay is driven by a top-down generative model, the predictive coding network successfully developed representations of MNIST digits and generated both memory and generative replays. The paper also studied the impact of the size of the hidden layer.

Overall, this paper presented some interesting experiments and discoveries of PCN in modeling memory replay, but failed to address its relevant literature and lacked analytical/experimental depth.

**Strengths:**

I found the class-conditioned generation from PCN's latent representations with multivariate Gaussian fitting interesting. Previous works (e.g., Oliviers et al. 2024) have discovered generative modeling with Langevin dynamics but may have missed this simple yet potentially effective approach to enable PCNs to generate unseen images from a latent distribution.

**Weaknesses:**

Despite the interesting discovery mentioned in strengths, this paper generally lacks both depth and width to be accepted as publication.

Problem with width:
- The authors have overlooked a considerable body of literature modeling memory in hippocampus using predictive coding networks (see below), which have largely covered 1) how memory and generative replay can be achieved in a PCN; 2) how memory can be stored both hierarchically and recurrently; 3) how the recurrent HPC and hierarchical EC can be combined in a single model. I strongly suggest the authors to carefully review this field and understand the proposed architecture's similarity and difference to these models
- The paper lacks a broader biological discussion on how the proposed architecture implies about the cortex e.g., is there evidence of error neurons in LGN, VC and EC? What experimentally verifiable predictions have this model made?

Problem with depth:
- The hierarchical structure of PCN has been extensively studied: in essence, the 2-layered model proposed here is no different from Rao and Ballard's (1999) original predictive coding model of visual cortex - does it mean LGN and VC alone can already perform memory replay? What is the role of multiple layers between LGN and EC if a single layer can already achieve replay/memory? Along this line, I'd suggest the authors to explore more the role of depth, rather than width, of PCNs e.g. in an 8-layer PCN, what are the respective roles of each intermediate layer?
- The experiments were only performed on MNIST. Today's PCNs can alreay have many layers and different architectures such as convolutional layers (see Pinchetti et al. 2025) to accommodate larger datasets such as ImageNet. Performing these experiments may not add value to the proposed model per se, but will definitely add value to its generalizability and applicability.
- The analysis of results are rather superficial. For example, Fig5a can benefit from a quantitative evaluation using FID scores of generations; the TSNE plots are not very informative as MNIST digits are known to be quite separable even in image space; some metrics of separability would help.

Other comments:
- Section 4 is basically a specific case of the equations in Section 3; instead of repeating Section 3, I'd describe in Section 4 how exactly the replay is achieved in mathematical terms.

References:
- Tang, Mufeng, et al. "Recurrent predictive coding models for associative memory employing covariance learning." PLoS computational biology 19.4 (2023): e1010719.
- Salvatori, Tommaso, et al. "Associative memories via predictive coding." Advances in neural information processing systems 34 (2021): 3874-3886.
- Salvatori, Tommaso, et al. "Learning on arbitrary graph topologies via predictive coding." Advances in neural information processing systems 35 (2022): 38232-38244.
- Tang, Mufeng, Helen Barron, and Rafal Bogacz. "Sequential memory with temporal predictive coding." Advances in neural information processing systems 36 (2023): 44341-44355.
-Oliviers, Gaspard, Rafal Bogacz, and Alexander Meulemans. "Learning probability distributions of sensory inputs with Monte Carlo predictive coding." PLOS Computational Biology 20.10 (2024): e1012532.

**Questions:**

See weaknesses.

---

> ### Author Response · Authors · 2025-11-19
>
> Problem with width:
> - The papers that you cited in relation to memory in PC all model auto-associative memory and not memory replay as in our paper. The model in Tang et al. is indeed similar to ours, combining a hierarchical PCN modeling the neocortex and a recurrent one-layer PCN modeling the hippocampus (actually only its area CA3), but they only focus on auto-associative memory. Let us recall the concepts of recall (or auto-associative memory) and replay, which are two different memory functions of the hippocampus. Recall is the pattern completion of corrupted inputs, whereas replay can be seen as the reinstantiation in the neocortex of episodic memories stored in the hippocampus (what we call "experience" replay) or as the generation of patterns in the neocortex based on knowledge stored in the hippocampus (what we call "generative replay"). When exposed to new sensory inputs, the hippocampus stores them in its synaptic connections, so that when it is presented with corrupted versions of these memories, it can complete (recall) them. During rest or sleep, the hippocampus replays memories (in "experience replay") or generates new patterns from stored memories (in "generative replay") for memory consolidation in the neocortex. In our paper, it is the second process of replay which we intend to model. To our knowledge, our work is the first to successfully model memory replay in a PCN, and to shed light on a trade-off between validation error and replay fidelity/separability when increasing the width of the top layer.    Additional experiments show that increasing the width of the top layer is also detrimental to the pattern completion of unseen patterns, suggesting an optimal width of 30. Furthermore, contrary to Tang et al., we think that the hippocampus cannot be fully modeled using PC because of its one-shot storage of memories, which is not compatible with PCNs because they gradually learn patterns through multiple epochs. That is why the top layer of our PCN corresponds to the entorhinal cortex, which is the input to the hippocampus, and not the hippocampus itself, which will be modeled in more detail in future work.
>
> Problem with depth:
> - We need at least three layers to model memory replay in a biologically plausible way. The top layer corresponds to the entorhinal cortex, which starts the replay as it is connected to the hippocampus. The middle layer converges to representations of the memory which can be decoded into pixel space thanks to their predictive nature. The bottom layer is the input layer, whose influence is canceled during memory replay, as the attention is focused on the top-down information flow. In practice, memory replay could be implemented with only two layers (VC and EC in particular) by initializing the bottom layer (VC) randomly and then letting the memory in EC be replayed in VC, but this model would not take into account the fact that the visual cortex can have structured sensory inputs during replay. We did not explore the role of depth because our focus is on modeling memory replay. In addition to memory replay, the visual cortex has other functions such as object detection, which could require multiple layers of processing, especially when working with more complex data than the MNIST dataset. The role of depth has been studied in other papers [1, 2].
> - The t-SNE plots were intended to show that the hierarchical representations of the patterns generated by conditioning on the class lie within the right clusters in each layer, which is not the case when the patterns are generated by sampling from the prior (not shown in the paper). The classification accuracy was introduced as a separability metric for the top layer, to investigate the influence of the layer width on separability, as previous work by Fontaine and Alexandre showed that the clusters overlapped in the top layer.
>
> [1] Brucklacher, M., Bohté, S. M., Mejias, J. F., & Pennartz, C. M. A. (2023). Local minimization of prediction errors drives learning of invariant object representations in a generative network model of visual perception. Frontiers in computational neuroscience, 17, 1207361. https://doi.org/10.3389/fncom.2023.1207361
>
> [2] T. Ed Li, Mufeng Tang, Rafal Bogacz; Predictive Coding Model Detects Novelty on Different Levels of Representation Hierarchy. Neural Comput 2025; 37 (8): 1373–1408. doi: https://doi.org/10.1162/neco_a_01769

---

> > ### Comment · Reviewer_siHR · 2025-11-19
> >
> > I agree with the authors on the conceptual difference between memory replay and recall. However, there are two main reasons that I, and possibly other reviewers, raised this concern on related literature:
> >
> > 1. The author did not cite these papers, and thoroughly discuss them within the context of this paper. I believe these papers are highly relevant, as they study the same region and related functions;
> > 2. It is unclear from the current paper, from a simulation point of view, what is the difference between memory recall and replay. I encourage the authors to think about the following question: if you were to perform memory recall with your model, what is the experimental process? How is it different from experience replay experiments? I'd then add more details to the experimental process to add clarity to the paper.
> >
> > Also, as reviewer wSYg pointed out, in addition to your response to reviewers, the authors should also consider editing the paper, or at least at this stage, point out in your replies how you will edit the paper in response to the comments. Also, the authors didn't provide an answer to my other concerns, for example testing the idea on other datasets, and more rigorous evaluation of generative replay results using e.g. FID scores. Considering these, I will keep my original score.

---

> > > ### Author Response · Authors · 2025-12-03
> > >
> > > 1. The references were added and discussed in the context of the paper in the Related works of the updated paper.
> > > 2. The experimental process of memory recall and replay has been added to the Methods section (section 4.3), including the equations.
> > >
> > > The model was not tested on other datasets in the current paper, because we thought that the MNIST dataset would be sufficient to test memory replay in PCNs. Of course, future work should extend the work to other datasets.
> > >
> > > We decided not to include FID scores for the generated images because we are not comparing the generation performance of our model to other models. The goal was to show that PCNs can support generative replay, by modelling hippocampal replay in an abstract way, rather than to evaluate the generative performance of PCNs. We expect the quality of the generated images to improve when the hippocampus will be modelled in more detail.

---

### Official Review · Reviewer_fWGk · 2025-10-30

**Soundness:** 3
**Presentation:** 3
**Contribution:** 1
**Rating:** 2
**Confidence:** 3

**Summary:**

The authors present a predictive coding model of episodic and generative replay involving the hippocampus and neocortex. The work spans a broad range of prominent ideas in predictive coding (Friston, Rao, Ballard etc), complementary learning systems & memory consolidation (McClelland etc), hippocampal replay, generative modelling such as VAEs and their application to memory consolidation (e.g. Spens & Burgess).

The highest level of the predictive coding hierarchy (containing the abstract latent states, MNIST digits in this paper) can be sampled from (as in VAE) to generate experiences - this is framed by the authors as performing the role of the hippocampus (which is not explicitly modelled in this work). From my understanding this differs from the role of HPC in systems consolidation models (e.g. Spens) where both HPC and neocortex receive sensory input x and each try to reconstruct the input x or predict a target y; in this way, hippocampal replays can provide the input/outputs for training the neocortical network. Here instead, the hippocampus is cast as generating a sample of neocortical (EC) activity itself i.e. the latents of the VAE.

The central result is that when activity memories are sampled at the top level (analogous to hippocampal replay), the network generates qualitatively recognisable samples from that class. Classes are separable clusters in each level when projected into low dimensions with t-SNE. More neurons in the highest latent layer increases the expressiveness of the abstract latent state and as a result the generated replays are more precise.

**Strengths:**

The mathematical description and diagram of the model are clear and easy to follow, as is the main text.

The work improves upon the result from Fontaine & Alexandre by demonstrating non-overlapping digit clusters in the latent spaces.

To my knowledge it is a novel idea to frame the top level of the PC hierarchy as a hippocampus, although the significance is not clear

**Weaknesses:**

I'm skeptical of the novelty of using PCNs to generate samples - they are generative models so it must follow that they can generate predictions, e.g. by clamping the top level. I also believe PCNs have been shown to accurately model MNIST digits. Unless I'm mistaken, the novel contribution then is relating the generation via sampling to hippocampus - however, beyond conception of this idea, the work in its current form does little to go beyond the (I believe) established fact that PCNs can generate predictions from sampling.

I believe it would also be helpful to relate back to memory consolidation, given that it is a core part of the story told in the abstract and introduction - in the current model, HPC cannot be used to train the neocortex since it does not generate inputs/outputs (as in Spens & Burgess) upon which cortex can be trained. If the model could be developed in such a way that these generative replays are functionally useful for training the PCN or downstream neocortical networks then it would be much more compelling. I believe one of the results of systems consolidation is that generalisable facts (mappings from x->y) get abstracted into neocortex but that irregular/incongruent facts are stored in hippocampus - perhaps in this PC model there would be interesting analyses of error neurons in these cases, or even that the error responses themselves are what get stored in hippocampus?

I feel in general it could benefit from a bit more 'meat' - the suggestions above are only suggestions but I do feel exploring the consolidation part further, or how predictive coding (via an extension to your modelling) can shed new light into consolidation - e.g. psychological phenomena, error driven responses etc. Perhaps a multimodal network could help? I think there might be much more compelling arguments you can make than focusing on the layer size!

**Questions:**

- How is replay error computed? Is it the reconstruction error in the top-down regime when the network predicts input using a previously seen latent sample?
- Second layer is just a linear layer - is it surprising that performance is the same with L=1 when the linear layer doesn't add any computational power?
- Why does validation error increase with larger layer 0? It seems to contradict the fact that classification
- Does the validation error trend in figure 6 contradict that of fig 3?
- "However, little work has been done in computational neuroscience to study how this theory can account for the functions
of the neocortex, in a biologically plausible way." - is this true? I was under the impression that predictive coding theory and modelling literature was extensive, focused on biological plausibility, and that PCNs have been used to model a variety of functions (though predominantly in the sensory domain).

---

> ### Author Response · Authors · 2025-11-19
>
> - As described in section 4.4, the replay error is the RMSE between the original and replayed images. The replayed image is obtained using the method described in section 4.3 (your description is correct).
> - With L=1, the model has only a linear layer (layer 0), but with L=2, we add a nonlinear layer (layer 0) to the linear layer (layer 1). Indeed, the layer index starts from the top of the hierarchy. We choose the bottom layer to be linear because of the efficient coding theory [1]. Therefore, showing that adding a nonlinear layer decreases reconstruction error was not obvious.
> - The results about the validation error can be interpreted in the light of efficient coding: any image can be described by the linear combination of a set of basis functions [1]. In our model, these basis functions correspond to the weights between the VC and LGN layers, which enable for the prediction of images in LGN. As we have tuned the number of neurons in VC to minimize the reconstruction error on the validation set (see Fig 6 in the appendix), we have found such a set of basis functions for the MNIST dataset. Therefore, adding a second layer (the entorhinal layer) can only decrease the validation error, even though the difference is not visible to the eye.
> - There is no contradiction between Fig 3 and 6, which correspond to the validation error when tuning the width of the top (EC) and bottom layer (VC) respectively. Fig 6 show that there is an intermediate VC width for which reconstruction of validation images is optimal, whereas Fig 3 shows that increasing the width of the EC layer decreases the reconstruction error. This can be explained by the previous argument about efficient coding.
> - There is indeed an extensive literature about predictive coding, but even though this model is biologically plausible in itself, it is often used in a machine learning setting as stated in the introduction. When functions of the neocortex are modeled by predictive coding in the literature (such as pattern completion or generation), they are not explicitly mapped to circuit mechanisms in the neocortex and the hippocampus.  In addition, the current work is to our knowledge the first to successfully model memory replay in a PCN. Therefore, the novelty of the work does not reside in the generation of samples, but in the modeling of the mechanisms underlying memory replay in the neocortex and the hippocampus. In particular, a central contribution of our work was to show that due to the trade-off between reconstruction error and separability/replay fidelity, the top layer must be big enough to ensure separability of the classes and accurate replay, but small enough to allow accurate reconstruction in the neocortex (in addition to better storage in the hippocampus). Additional experiments show that increasing the width of the top layer is also detrimental to the pattern completion of unseen patterns, suggesting an optimal width of 30.
>
> [1] Olshausen, B., Field, D. Emergence of simple-cell receptive field properties by learning a sparse code for natural images. Nature 381, 607–609 (1996). https://doi.org/10.1038/381607a0

---

### Official Review · Reviewer_wSYg · 2025-10-30

**Soundness:** 3
**Presentation:** 4
**Contribution:** 2
**Rating:** 6
**Confidence:** 5

**Summary:**

This paper presents a computational model of memory replay, framing it as a computational process based on predictive coding (PC). This work uses a 3-level hierarchical predictive coding network (PCN) trained on the full MNIST dataset. The model's layers are explicitly mapped to a simplified visual hierarchy: the input layer (level 2) is mapped to the LGN, the hidden layer (level 1) to the visual cortex (VC), and the top layer (level 0) to the entorhinal cortex (EC).The paper demonstrates two distinct forms of memory replay: (a) Experience Replay: This involves clamping a previously stored latent representation (from an item in the training set) onto the top EC layer. To allow this top-down signal to generate a percept, the precision of the input LGN layer is set to zero. This gates the network from bottom-up sensory input and allows the top-down prediction to flow down, generating a replayed image at the input. (b) Generative Replay: This is presented as a novel form of replay. The authors fit a class-conditioned multivariate Gaussian distribution to the entire latent space of the EC layer (using all stored representations from the training set). By sampling new latent codes from this Gaussian, they can then use the same top-down replay mechanism to generate novel, unseen exemplars of the digits.The central and most interesting finding of the paper is a non-obvious trade-off: increasing the number of hidden units in the top (EC) layer ($n_0$) worsens the network's ability to reconstruct novel inputs (the validation error increases). However, this same change improves the quality of its replayed images (the replay error decreases) and simultaneously improves the linear separability of the latent representations, as measured by a simple logistic regression classifier.

**Strengths:**

The paper does an excellent job explaining PC and mapping it clearly to a specific cognitive function (memory replay) and its supposed neuro-anatomical substrate (LGN, VC, EC). The distinction between the "perception" (encoding) and "replay" (decoding/generative) pathways is elegant. The explanation of the mechanism for replay—clamping the top layer and "gating" the bottom layer by setting its precision ($\Sigma_2^{-1}$) to zero—is a biologically-plausible hypothesis for empirical phenomenon observed in the literature.

The central discovery of a trade-off between reconstruction and replay quality/separability is the most valuable part of this paper in my opinion. The finding that increasing top-level units hurts reconstruction while helping replay and separability suggests a functional specialization in a hierarchy: lower-level layers may be optimized for precise fidelity, while higher-level layers are optimized for abstract, separable representations that are good for replay and classification. This finding, in isolation, is a useful contribution to the learning representation community, as it provides a concrete example of how a hierarchical system might resolve the tension between detailed reconstruction and abstract categorization.

The work is well-grounded in neuro-inspired principlesn and relevant empirical observations. It provides a mechanistic, more biologically plausible account of generative memory replay that does not rely on biologically implausible mechanisms like backpropagation. It connects a known cognitive function (replay) to a specific, testable neural mechanism (top-down generation, precision-gated).

**Weaknesses:**

Although the results stated in strength are interesting, the authors could have done a much better job framing their contribution wrt the literature, as some of the interpretations in the submission is directly related to, if not identifical with prior, unreference works. Here are a few critical omissions:

1. https://arxiv.org/abs/2109.08063. This work is quite critical as it's one of the earliest works to connect memory/replay to PC with a computational model & experiments on CV benchmarks on MNIST. It also makes the connection of PC to the hippocampus as a "memory index and generative model".

2. https://philpapers.org/rec/EDLPPC. This thesis explicitly reconstructs the overarching conceptual argument this paper relies on: that PC is a "unifying theory of perception, imagination, memory, and dreaming," implemented across the brain's hierarchies. It explains imagination as "offline perception" and dreaming as "unconstrained imagination" by modulating sensory input/precision—the exact mechanism this paper uses for replay. Although the computational model is less sophisticated than the one presented in the submission, it is crucial to ground the discussion by properly attributing the intellectual lineage of the idea that PC could be a unified theory for various mental functions.

3. https://direct.mit.edu/neco/article/37/8/1373/131383/Predictive-Coding-Model-Detects-Novelty-on. This paper follows the same theme of hierarchical abstraction as the current submission. It uses PCNs to show that different layers detect novelty at different levels of abstraction: "sensory (pixel) novelty" in low layers versus "semantic (digit) novelty" in high layers. This directly overlaps with the current submission's theme of finding separable representations in the top (EC) layer. Moreover, the idea that different brain hierarchy/areas can correspond to different layers in a hierarchical PCN is already fleshed out in that paper, which the current submission claims without references.

**Questions:**

My biggest question would be how would your contextualize the findings and novelty wrt the literature such as discussed in weaknesses. Also, the core finding of the trade-off in Figure 3 is interesting and what is your hypothesis for why this occurs? Why does a wider top-level layer ($n_0$) hurt reconstruction (increase validation error) while improving replay and separability? Does this imply that the top layer is being forced to learn more abstract, compressed representations that discard instance-specific variance (which hurts reconstruction) but better capture the class-defining essence (which aids separability and replay)? I'm currently on the fence regarding accpet/reject of this paper and may be swayed if my questions are addressed sufficiently.

---

> ### Author Response · Authors · 2025-11-19
>
> Auto-associative memory and image generation with PCNs have been studied in a machine learning setting (see Pinchetti et al. which evaluates different PCN models on ML tasks, including the model of Salvatori et al.), but the relation to the brain, especially the hippocampus, has been overlooked. Although Salvatori et al. and later Tang et al. [1] proposed that the top layer of their PCN model could be the hippocampus, they only model auto-associative memory (hippocampal recall) and not memory replay, as in our paper. Furthermore, contrary to authors of Salvatori et al. and Tang et al., we think that the hippocampus cannot be fully modeled using PC because of its one-shot storage of memories, which is not compatible with PCNs because they gradually learn the patterns through multiple epochs. That is why the top layer of our PCN corresponds to the entorhinal cortex, which is the input to the hippocampus, and not the hippocampus itself, which will be modeled in more detail in future work. Indeed, our paper is not the first to map the hierarchical PCN to different brain areas or to rely on the idea that PC is a "unifying theory of perception, imagination, memory, and dreaming", but to our knowledge it is the first to successfully implement memory replay in such a framework and to study the effect of the width of the top layer.
>
> The positive influence of layer width on separability and replay can be understood as follows. It is known that expanding the dimensionality of the activity space of patterns increases their linear separability [2]. We hypothesize that this increased separability leads to better replay, as there is less interference between patterns. Another way to see it would be in terms of compression: as we are compressing the pixel space into a lower dimensional latent space (the entorhinal layer), information can be lost, resulting in replays with lower fidelity as the dimensionality is decreased.
>
> The results about the validation error can be interpreted in the light of efficient coding: any image can be described by the linear combination of a set of basis functions [3]. In our model, these basis functions correspond to the weights between the VC and LGN layers, which enable for the prediction of images in LGN. As we have tuned the number of neurons in VC to minimize the reconstruction error on the validation set (see Fig 6 in the appendix), we have found such a set of basis functions for the MNIST dataset. Therefore, adding a second layer (the entorhinal layer) can only decrease the validation error, even though the difference is not visible to the eye.
>
> I agree with your interpretation of the implications of the results. Overall, the results suggest that the hippocampus stores compressed, separable representations of patterns that can be replayed in the neocortex. Due to the trade-off between reconstruction error and separability/replay fidelity, the top layer must be big enough to ensure separability of classes and accurate replay, but small enough to allow accurate reconstruction in the neocortex (in addition to better storage in the hippocampus). Additional experiments show that increasing the width of the top layer is also detrimental to the pattern completion of unseen patterns, suggesting an optimal width of 30.
>
> [1] Tang, Mufeng, et al. "Recurrent predictive coding models for associative memory employing covariance learning." PLoS computational biology 19.4 (2023): e1010719.
>
> [2] Cayco-Gajic, N. A., & Silver, R. A. (2019). Re-evaluating Circuit Mechanisms Underlying Pattern Separation. Neuron, 101(4), 584–602. https://doi.org/10.1016/j.neuron.2019.01.044
>
> [3] Olshausen, B., Field, D. Emergence of simple-cell receptive field properties by learning a sparse code for natural images. Nature 381, 607–609 (1996). https://doi.org/10.1038/381607a0

---

> > ### Comment · Reviewer_wSYg · 2025-11-19
> >
> > I tend to agree with the authors on the interpretations provided. However, it is crucial to contextualize their findings against directly related, previous works /textit{inside} your submission (ideally in the related works section), not just in responses to reviewer comments. The core ideas/arguments of the authors' submissions are still closely related to the references I mentioned in my review. For instance, even the authors admitted in their reply:
> >
> > 'Indeed, our paper is not the first to map the hierarchical PCN to different brain areas (reviewer's note: first mentioned in [1] and extended in greater length in [3]) or to rely on the idea that PC is a "unifying theory of perception, imagination, memory, and dreaming" (reviewer's note: this is the main argument in [2] and extensively discussed there), but to our knowledge it is the first to successfully implement memory replay in such a framework and to study the effect of the width of the top layer'
> >
> > I remained to be on the fence regarding accept/reject on this work, but I think properly contextualizing your work within the literature should be and is a bare minimum for ICLR submissions.
> >
> > [1] https://arxiv.org/abs/2109.08063,
> > [2] https://philpapers.org/rec/EDLPPC,
> > [3] https://direct.mit.edu/neco/article/37/8/1373/131383/Predictive-Coding-Model-Detects-Novelty-on

---

> > > ### Author Response · Authors · 2025-12-03
> > >
> > > The references you pointed out have been added to the Related works, Methods and Discussion sections of the updated paper. The interpretations provided were also added to the Discussion section.

---

### Official Review · Reviewer_YYAe · 2025-10-30

**Soundness:** 2
**Presentation:** 2
**Contribution:** 1
**Rating:** 2
**Confidence:** 5

**Summary:**

This paper proposes a biologically motivated predictive coding model to simulate memory replay and generative imagination in hippocampo-neocortical interactions. Using the traditional predictive coding frameworks, it studies both episodic replay of stored memories and generative replay to simulate imagination. They provide detailed quantitative and qualitative evaluations on MNIST, analyzing reconstruction accuracy, replay quality, and latent class separability, offering insights about how network architecture impacts replay dynamics and memory consolidation processes.

**Strengths:**

The paper aligns predictive coding architectures with biologically realistic neural circuitry from the Rao and Ballard algorithm (paper not cited!). It distinguishes episodic replay (recall) from generative replay (imagination), contributing to emerging efforts to model both memory consolidation and creative generation within a unified predictive coding system.

**Weaknesses:**

There is little novelty in this work: there is a large body of literature that tests predictive coding networks on associative memory problems. First, from the more "ML" side of image storage and retrival, there is a work that uses the same framework you propose, but goes beyond the MNIST experiment you performed, all the way to ImageNet [1]. Then, there are multiple extensions: the first uses a categorical prior on the last layer to provide the model with exponential capacity [2], another one, more focused on the neurosciences, maps this model to different brain regions, developing a theory of how recurrent PCNs can be attractor points [3]; a third one updates the original algorithm to provide the PCN with a write-erase system that allows you to free the memory of datapoints you are not interested in [4]. There are also extensions on the use of PCNs for temporal data [5], and grid cells [6].

[1] Salvatori, Tommaso, et al. "Associative memories via predictive coding." Advances in neural information processing systems 34 (2021): 3874-3886.

[3] Tang, Mufeng, et al. "Recurrent predictive coding models for associative memory employing covariance learning." PLoS computational biology 19.4 (2023): e1010719.

[4] Yoo, Jinsoo, and Frank Wood. "Bayespcn: A continually learnable predictive coding associative memory." Advances in Neural Information Processing Systems 35 (2022): 29903-29914.

[5] Tang, Mufeng, Helen Barron, and Rafal Bogacz. "Sequential memory with temporal predictive coding." Advances in neural information processing systems 36 (2023): 44341-44355.

[6] Tang, Mufeng, Helen Barron, and Rafal Bogacz. "Learning grid cells by predictive coding." arXiv preprint arXiv:2410.01022 (2024).

**Questions:**

How does your model differ from that of Salvatori et al.?

---

> ### Author Response · Authors · 2025-11-19
>
> Our model and the model from Salvatori et al. are based on the same model by Friston, who adapted the model of Rao and Ballard to a variational inference framework and extended it to $n$ layers to map it to the cortical hierarchy. However, our model is not designed to test associative memory or recall, i.e. the pattern completion of a corrupted input, in PCNs as in Salvatori et al. or Tang et al. (and most of the other papers you cited). Instead, our work models memory replay, as the reinstantiation in the neocortex of episodic memories stored in the hippocampus or as the generation of patterns in the neocortex based on knowledge stored in the hippocampus, and highlights the importance of the choice of the size of the entorhinal layer. Whereas recall intends to complete the pattern in the input layer, replay cancels the influence of the input to attend only to top-down information given by the hippocampus. Contrary to authors of Salvatori et al. and Tang et al., we think that the hippocampus cannot be fully modeled using PC because of its one-shot storage of memories, which is not compatible with PCNs because they gradually learn the patterns through multiple epochs. That is why the top layer of our PCN corresponds to the entorhinal cortex, which is the input to the hippocampus, and not the hippocampus itself, which will be modeled in more detail in future work. Although the submitted paper focuses on memory replay, additional experiments show that increasing the width of the top layer in our model is detrimental to the pattern completion of unseen patterns, suggesting an optimal width of 30.

---

> > ### Comment · Reviewer_YYAe · 2025-11-19
> >
> > I see your point, and I got the differences you mention between the associative memory papers mentioned above, and yours. However, I still do not believe there is enough novelty in this work for an ICLR submission: What you are doing is observe a datapoint (your MNIST image); compute the posterior;  re-generate the original image using the compressed representation. Isn't this is the standard way of using predictive coding networks? I see your point that it slightly differs from Salvatori et al., mostly in the way you are using test points and not training points during the evaluation, but then how is this different from standard variational inference?
> >
> > My concerns are:
> > 1) The architecture and training algorithms are not novel (Same as Rao&Ballard, Friston, as you state);
> > 2) The evaluation is not novel (what you are proposing is the standard way of testing such models);
> > 3) The study is a simple curve over a single hyperparameter (validation error on a different number of latent states)

---

> ### Author Response · Authors · 2025-11-20
>
> Memory replay happens during rest or sleep. When it happens during rest, the brain can receive sensory input during replay. When this happens, the brain detaches from its current senses to attend to the memory replay. As stated in the end of section 3, according to Friston, attention could be mediated by the precision matrix (or inverse covariance matrix) in PCNs: lower precision in the input layer will correspond to less attention to the input. Therefore, in our model, we set the precision matrix of the input layer to $\mathbf{0}$ during memory replay, so that the input does not influence the representations in the visual cortex during replay: in this way, attention is focused on the replay and not on the input. The standard way of using predictive coding networks is to set the covariance matrices to $\mathbf{I}$ or to learn them.
>
> What we are doing in the evaluation is not simply replacing training points by test points. The current version of the paper shows a trade-off between the reconstruction error (i.e. the RMSE between the original and reconstructed images, and not the recalled images as in Salvatori et al.) on the validation set, and the replay fidelity and separability of classes in the top layer, as we increase the width of the top layer. New experiments, which will be added in the paper, show that the optimal width can be chosen by looking at the pattern completion performance of the models, evaluated on the validation set, which decreases with layer width. The same metric as in Salvatori et al. could be used (and in this case, it is just replacing the training points by test points).  Therefore, the quantitative study is a combination of curves over the layer width, complemented with a qualitative study. The influence of widths in PCNs has not been studied before, because previous works focused on perception and auto-associative memory in particular.

---

### Meta-Review · Area_Chair_8uLW · 2025-12-19

**Summary:**

This submission presents a biologically motivated predictive coding model of hippocampo–neocortical interactions, focusing on memory replay rather than associative recall. The authors implement a hierarchical predictive coding network mapped onto LGN, visual cortex, and entorhinal cortex, and model replay via top–down clamping of the entorhinal representation combined with precision gating of sensory input. The paper studies both experience replay and generative replay on MNIST, and its main empirical finding is a trade-off controlled by the width of the top layer: increasing capacity improves replay fidelity and class separability but degrades reconstruction and pattern completion on unseen inputs. The model is technically sound, clearly described, and carefully evaluated. However, after considering the reviews and the overall novelty threshold for ICLR, the decision is to reject, with encouragement to resubmit to a more suitable venue or after strengthening the contribution.

**Reviewer Concerns:**

All reviewers agree that the paper is technically correct, well presented, and grounded in established predictive coding theory. There is consensus that the implementation is faithful to known PCN formulations and that the reported results are internally consistent. Where the reviews diverge is in their assessment of novelty and impact.

One reviewer takes a strongly negative stance, arguing that the architecture, inference rules, and evaluation protocol closely follow prior predictive coding work, particularly associative memory models, and that replay via top–down generation is essentially standard use of a generative hierarchy. From this perspective, the empirical study is seen as limited to MNIST and to a single hyperparameter sweep, and therefore insufficiently novel for ICLR.

A second reviewer is more balanced and identifies the core trade-off between reconstruction and replay quality as the most interesting contribution of the paper. This reviewer finds the biological interpretation of replay through precision gating to be elegant and potentially valuable, but expresses concern that the submission initially under-contextualized its relationship to prior work on predictive coding, imagination, and replay. Even after the authors’ clarifications, this reviewer remains on the fence, noting that better positioning within the literature is necessary and that the contribution, while real, may be incremental.

The third reviewer is also negative overall, questioning whether the ability of predictive coding networks to generate samples when clamping higher-level latents is surprising, and expressing skepticism about the functional role of replay in the current model, since it is not used for consolidation or further learning. This reviewer views the focus on layer width as somewhat arbitrary and would have preferred a broader or more functionally grounded demonstration.

Taken together, the reviews indicate agreement on soundness and clarity, but insufficient consensus on the level of novelty and impact required for acceptance at ICLR. With two reviewers recommending rejection and the most positive reviewer indicating only marginal support and openness to rejection, the balance of evidence supports a reject decision. The work is viewed as a solid and thoughtful contribution that would likely be better received in a venue emphasizing mechanistic modeling and computational neuroscience, or after extension to broader empirical settings or clearer differentiation from closely related predictive coding literature.

**Reviewer Scores:**

Based on the tone  and content of the discussion, the follow-up comments, and the authors’ responses, my assessment is that the reviewers’ scores would have changed little overall, though their confidence in those scores may have shifted slightly.

Reviewer YYAe would almost certainly have maintained the same overall score and reject recommendation. While this reviewer acknowledged understanding the authors’ clarifications about replay versus recall and the role of precision gating, their core concern was about novelty relative to existing predictive coding and associative memory literature. The subsequent discussion did not address this concern to their satisfaction, and the reviewer explicitly reiterated that they did not see enough novelty for an ICLR submission. At most, their confidence might have been marginally reinforced rather than softened.

Reviewer wSYg might have slightly increased their confidence in a marginal score but not clearly moved to a strong accept. This reviewer found the central trade-off result interesting and agreed with the authors’ interpretations in the discussion, and their questions were answered in a technically reasonable way. However, even after acknowledging these points, they continued to emphasize that the contribution needed better contextualization within prior work and that novelty remained borderline. If fully involved in the discussion, this reviewer might have remained at roughly the same score (e.g., marginal accept / borderline), perhaps leaning a bit more positive in terms of soundness and clarity, but still not strongly advocating acceptance.

Reviewer fWGk would likely have kept a reject score, though possibly with slightly reduced uncertainty. Their main reservations concerned the conceptual payoff and perceived obviousness of generative replay in predictive coding networks, as well as the lack of a direct link to consolidation or learning benefits. While the authors’ responses clarified technical details and biological motivation, they did not fundamentally change the scope of the contribution in a way that addresses this reviewer’s higher-level concerns. As a result, the score would probably remain negative, with similar or slightly higher confidence.

---

### Decision · Program_Chairs · 2026-01-26

Reject